# Divergent resistance pathways amongst SARS-CoV-2 PLpro inhibitors highlight the need for scaffold diversity

Xinyu Wu[1,2], Shane M. Devine[1,2], Margareta Go[1], Julie V. Nguyen[1], Bernadine G. C. Lu[1,2], Katie Loi[1,2], Nathan W. Kuchel[1,2], Kym N. Lowes[1,2], Jeffrey P. Mitchell[1,2], Guillaume Lessene[1,2,3], David Komander[1,2], Matthew E. Call[1,2]*, Melissa J. Call[1,2]*

**1** The Walter and Eliza Hall Institute of Medical Research, Parkville, Victoria, Australia, **2** Department of Medical Biology, University of Melbourne, Parkville, Victoria, Australia, **3** Department of Biochemistry and Pharmacology, University of Melbourne, Parkville, Victoria, Australia

* mecall@wehi.edu.au (MEC); mjcall@wehi.edu.au (MJC)

## Abstract

Drug-escape, where a target evolves to escape inhibition from a drug, has the potential to lead to cross-resistance where drugs that are structurally related or share similar binding mechanisms all become less effective. PLpro inhibitors are currently under development and many emerging PLpro inhibitors are derived from GRL0617, a repurposed SARS-CoV PLpro inhibitor with moderate activity against SARS-CoV-2. Two leading derivatives, PF-07957472 and Jun12682, demonstrate low nanomolar activity and display activity in mice. WEHI-P8 is structurally distinct but binds to a similar pocket adjacent to the active site as GRL0617-like compounds. Using deep mutational scanning, we assessed the potential for PLpro to develop resistance to PF-07957472, Jun12682, and WEHI-P8. PF-07957472 and Jun12682 exhibited largely overlapping escape mutations due to their shared scaffold and binding modes, whereas WEHI-P8 resistance mutations were distinct. These findings underscore the importance of developing structurally diverse inhibitors to minimize resistance risks and ensure that viral mutations against one compound do not compromise the efficacy of others.

## Author summary

GRL0617, originally developed for SARS-CoV, was quickly identified as an inhibitor of SARS-CoV-2 PLpro and has since become a widely used scaffold for developing more potent and drug-like inhibitors. In this study, we compared two GRL0617-based compounds with a third, structurally unrelated inhibitor identified through high-throughput screening. Using deep mutational scanning, a method that tests thousands of possible changes to the viral enzyme, we evaluated how each compound might be affected by mutations the virus could develop to resist

**Data availability statement:** Raw data is submitted on MaveDB (urn:mavedb:00001248). Raw and additional data can also be found in the supplementary material. All genome sequences and associated metadata in this dataset are published in GISAID's EpiCoV database under GISAID Identifier: EPI_SET_250801st. To view the contributors of each individual sequence with details such as accession number, Virus name, Collection date, Originating Lab and Submitting Lab and the list of Authors, visit EPI_SET_250801st. DOI: https://doi.org/10.55876/gis8.250801st. EPI_SET_250801st is composed of 17,113,940 individual genome sequences. The collection dates range from 2010-12-06 to 2025-07-09; Data were collected in 222 countries and territories.

**Funding:** This work was supported by: a Wellcome Trust Innovator Award 222698/Z/21/Z to DK, GL, and MJC, MRFF grants MRF2002119 to DK and GL, and MRF2016781 to DK, GL, and MJC; and salary support from NHMRC Investigator Grants (GNT2016461 to GL, GNT1178122 to DK). The funders had no role in study design, data collection and analysis, decision to publish, or preparation of the manuscript.

**Competing interests:** I have read the journal's policy and the authors of this manuscript have the following competing interests: DK is founder, shareholder and SAB member of Entact Bio and Proxima Bio. WEHI-P8 is protected under provisional patent AU2024900559. The authors XW, SMD, BGCL, KL, NWK, KNL, JPM, GL, DK and MJC declare a competing interest regarding the development of WEHI-P8.

treatment. We found that the two GRL0617-based compounds were vulnerable to similar resistance mutations, suggesting that if the virus becomes resistant to one, it may also resist the other. The third inhibitor, however, was sensitive to a different set of mutations, many of which we predict will also reduce PLpro activity. These findings provide insight into how drug resistance might evolve and offer a framework for designing more robust antiviral treatments.

## Introduction

The COVID-19 pandemic caused by Severe Acute Respiratory Syndrome Coronavirus 2 (SARS-CoV-2), created havoc around the globe. Even six years after its initial outbreak in 2019, individuals continue to suffer from infections, with ongoing loss of life. In addition to acute disease, some individuals suffer from Post-Acute Sequelae of COVID-19 (PASC) [1–3], also known as long COVID with symptoms persisting months to years after recovery from the initial infection. The severity of infections and the persistence of symptoms continue to drive the development of SARS-CoV-2 therapeutics, particularly viral protease inhibitors, due to their low cost and oral bioavailability [4–7].

The SARS-CoV-2 genome encodes 29 proteins, of which 16 are non-structural proteins (Nsps) located on two polyprotein chains, pp1a and pp1ab [8]. These Nsps, which form the viral replication machinery, are released through proteolytic cleavage of the polyprotein chains by the proteases Main protease (Mpro) and Papain-Like protease (PLpro). PLpro cleaves defined motifs at the junctions between Nsp1, Nsp2, Nsp3, and Nsp4, while Mpro cleavage is required for liberation of the remaining Nsps. Existing therapeutics, Paxlovid (a combination of Mpro inhibitor nirmatrelvir [4] and the CYP3A4 inhibitor ritonavir [9]) and ensitrelvir [10], target Mpro, with PLpro inhibitors yet to reach the market. While the role of Mpro is focused on polypeptide cleavage, PLpro also plays a role in deconjugating ubiquitin and ubiquitin-like molecules, which are important for cellular innate immunity [11,12]. Furthermore, a recent study demonstrated that PLpro can directly activate sensory neurons, initiating pain and airway reflexes [13]. Thus, targeting PLpro instead of or in addition to Mpro during SARS-CoV-2 infection may have additional benefits.

PLpro is a globular domain of Nsp3 formed from ubiquitin-like (Ubl2), thumb, palm, and finger domains (S1A Fig) and retains proteolytic activity even when expressed in isolation of surrounding Nsp3 sequences. Engagement of the minimal cleavage motif $Leu_{P4}X_{P3}Gly_{P2}Gly_{P1} \downarrow$ is mediated by a deep S4 pocket that buries the Leu at the P4 position with the remaining sequence accommodated through the shallow S3 pocket and via a narrow channel accommodating the P2 & P1 glycines that connects to the active site (S1 Fig). Both the binding pockets and the narrow channel that accommodates substrates are flanked on one side by residues found along a flexible β-hairpin, termed the blocking loop. The active site, where enzymatic reactions occur, is situated on a flat surface that spans the thumb and palm domains and due to these structural constraints, inhibitor design has focused on blocking the deeper substrate binding S4 pocket instead of binding to the active site directly [14–17].

The SARS-CoV outbreak in 2002 and subsequent identification of MERS in 2012 highlighted the need for coronaviral inhibitors. Two PLpro inhibitor series, GRL0617 [16] and 3k/5c [17], were identified prior to the SARS-CoV-2 pandemic, and these have proven useful starting points for the development of high potency PLpro inhibitors. The GRL0617 series has gained the most traction and both engages the blocking loop and occupies the S4 pocket to inhibit PLpro activity [16] (S2 Fig). GRL0617 itself had single-digit micromolar activity when first screened against SARS-CoV-2 PLpro using purified protein in biochemical assays and an EC50 of over 10 µM in cellular assays [16,18]. While not potent enough for immediate clinical use, it provided a starting point for further optimization to enhance its efficacy [5,6,19–22], with medicinal chemistry efforts focused on improving fit in the S4 and S3 pockets, promoting steric hinderance of ubiquitin binding [6,23,24] and improving hydrogen-bonding networks from neighboring residues. Recently two inhibitors of this class, PF-07957472 [19] and Jun12682 [6] (Fig 1A) demonstrated substantial effectiveness in infected mouse models. Another example, HL-21, has entered phase II clinical trials in China [25]. Most recently, a third, chemically distinct, scaffold has been reported, the WEHI-P series [26], developed from a high-throughput screening hit and subsequent medicinal chemistry campaign against SARS-CoV-2 PLpro. Crystallization of WEHI-P4 with PLpro (Fig 1A) demonstrated a distinct binding mode compared to both GRL0617 and 3k/5c series (S2 Fig), including a reduced requirement for the blocking loop. WEHI-P8 was subsequently designed with improved activity.

In our previous work, we identified drug escape variants of the 3k/5c series [14,17] using a Deep Mutational Scanning (DMS) [18] workflow, measuring the activities of nearly 6,300 single-residue substituted PLpro variants in a cellular FRET assay with and without inhibitor treatment; those maintaining activity in the presence of 3k and 5c were classified as drug escape variants. Here, we investigated the escape profiles of GRL0617-like compounds and the WEHI-P scaffold. We examined two GRL0617-like compounds, PF-07957472 and Jun12682, along with one WEHI-P compound, WEHI-P8, using a refined DMS workflow. Our findings reveal hotspots for drug escape, with at least one variant at each hot spot validated via recombinant protein assays. GRL0617-like compounds had similar escape variants that were largely distinct from variants that enable escape from WEHI-P8. In the context of our previous results identifying 3k/5c escape variants, this study offers critical insights into the potential drug resistance patterns for each PLpro inhibitor scaffold, establishing a prospective framework for monitoring variants that arise as these inhibitors enter clinical use and for adapting therapeutic strategies in response to emerging resistance.

## Results

### Inhibitors are active in a cellular FRET assay

We previously established a cellular FRET assay to measure PLpro proteolytic activity [18]. This assay relies on a PLpro sensitive biosensor comprising a mClover3 donor fluorophore fused to a mRuby3 acceptor fluorophore via a PLpro cleavable linker – TLKGGAPTKV [18]. Cells expressing the biosensor lose FRET upon PLpro expression (Fig 1B). This assay enabled the examination of the cellular effectiveness of PLpro inhibitors [18], including GRL0617 [16], Jun9-84-3 [27], XR8–89 [28], 3k [17], and 5c [14,15,17] at blocking PLpro's enzymatic activity. We utilized the same assay to profile the dose-response relationships for three new compounds: PF-07957472 [19], Jun12682 [6], and WEHI-P8 [26]. An 8-point titration was performed, and inhibition of PLpro's activity was recorded as the percentage of FRET positive signal (FACs gating strategy in S3 Fig). All three compounds exhibited sub-micromolar efficacy, with EC50 values of 0.10 µM for WEHI-P8, 0.11 µM for PF-07957472, and 0.74 µM for Jun12682 (Fig 1C). Literature values showing the efficacy of these inhibitors in cell-based assays measuring viral replication are similar: PF-07957472 at 0.147 µM [19]; WEHI-P8 at 0.298 µM [26]; Jun12682 at 0.42 µM [6].

### Identification of escape variants

To determine PLpro variants that can escape each inhibitor, we assayed nearly all (>99%) single-site substitutions of PLpro in a series of DMS experiments that determined PLpro activity in the presence of each drug, and baseline PLpro

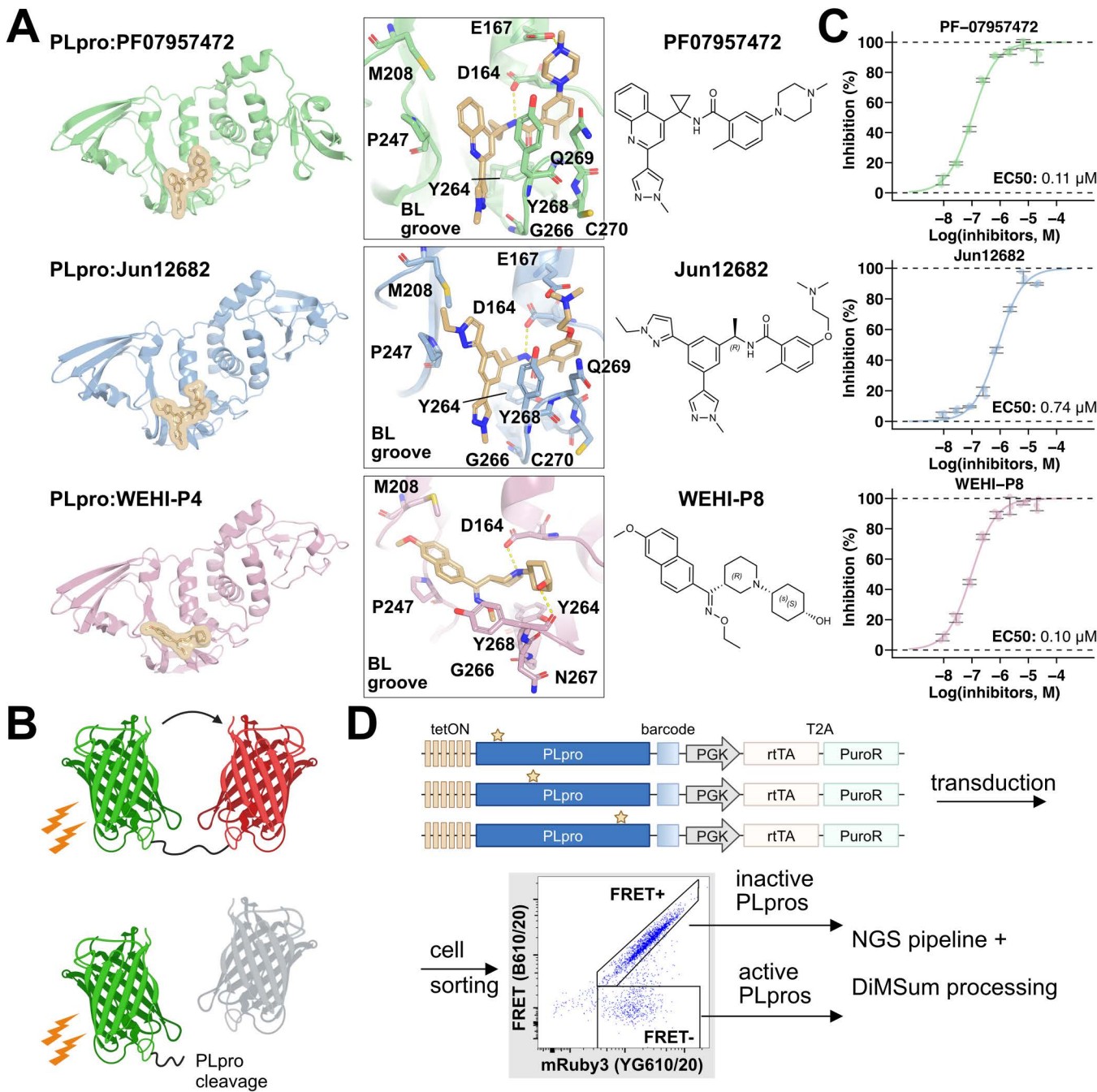

**Fig 1. PLpro inhibitors, cellular assay and DMS workflow.** (**A**) Structures of PLpro bound to the inhibitors (in yellow). From top to bottom: PF-07957472 (PDB: 9CSY, PLpro in green) [19], Jun12682 (PDB: 8UOB, PLpro in blue) [6], and WEHI-P4 (9CYD, PLpro in red) [26]. The central panel zooms in on these compounds and their surroundings, highlighting key interaction residues. Chemical structure formulae are shown in the right panel. (**B**) Schematic of the FRET biosensor design [18], which links mClover3 to mRuby3 via a PLpro cleavage motif. Upon excitation of mClover3, energy is transferred to mRuby3, producing a FRET signal; however, cleavage of the linker by PLpro disrupts this transfer, resulting in diminished signal. Created in BioRender. Wu, X. (2025) https://BioRender.com/aopcy0t (**C**) Dose-response curves showing the cellular efficacy of PF-07957472 (green, top), Jun12682 (blue, middle), and WEHI-P8 (red, bottom). The x-axis represents the log-transformed inhibitor concentration (unit: M), while the y-axis represents the normalized FRET signals. The signals from no PLpro treatment and no inhibitor treatment are normalized to 100 and 0 respectively. The plots are representative experiment of two biological replicates, while reported EC50 values represent the averages obtained from two biological replicates. Error bar: mean±SD. (**D**) The DMS workflow. PLpro variants controlled by tet-on and containing a barcode after the stop codon are transduced into 293T cells and sorted based on FRET signal. These populations are harvested for genetic material, which is sequenced by Illumina and scored using DiMSum [29].

activity in our dox-inducible system, and compared each dataset with previously established PLpro activity data [18]. Our PLpro library comprises a pool of barcoded single-residue-substituted variants with 99% coverage of PLpro residues 1–315, which correspond to 746–1060 in full-length Nsp3 (6263 of 6301 theoretical variants), installed in a lentiviral backbone that enabled dox-controlled expression of PLpro and puromycin selection of transduced cells (Fig 1D).

We transduced this library into HEK293T cells that stably expressed our PLpro biosensor at low MOI in four independent replicates. We transduced enough cells to ensure each variant was represented at least 10-fold in the population. Cells were concurrently treated with dox to induce PLpro expression and with inhibitors at their respective EC95 concentrations (3 µM for PF-07957472 and WEHI-P8; 10 µM for Jun12682). The next day, we sorted the cells into FRET$^+$ and FRET$^-$ gates and prepared genetic material from total RNA for next-generation sequencing (NGS). We similarly sorted cells without dox treatment to measure baseline PLpro activity driven by leaky expression in our system. After sorting, these cells were treated with dox for 4 h to induce mRNA prior to harvest. Reverse-transcribed RNA from cells was used to amplify barcodes that tagged each variant and were located downstream of the PLpro coding region stop codon (Fig 1D). These amplicons were sequenced by Illumina and used to determine variant frequency in each cellular population. Fitness scores and their associated errors were calculated with DiMSum [29], and these reported on the enrichment of variants in the FRET$^-$ gate (S1 Dataset **-** Raw Data).

We first inspected fitness scores calculated from cells that had been treated with inhibitors. We expect that most PLpro variants will remain sensitive to each inhibitor and remain in the FRET+ gate, indicating that the biosensor has not been cleaved. In contrast, variants bearing mutations that allow PLpro activity and prevent inhibitor binding will cleave the biosensor and fall in the FRET$^-$ gate, displaying enrichment and returning higher DiMSum fitness scores. While perfect inhibition should result in both wildtype and nonsense variants giving similar fitness scores (because both should fall in the FRET$^+$ gate), we nevertheless saw evidence of residual PLpro activity, indicating inhibition was not complete. This allowed us to normalize fitness scores for the wildtype and the mean of nonsense variants (to aa305) to 1 and 0, respectively, to facilitate comparison of each dataset (S4–S6 Figs, S1 Dataset - Normalized fitness and SE).

Sequence-function maps (S4–S6 Figs) showed mutations in several regions had higher than wildtype activity scores in the presence of inhibitors, particularly those at D164, E167, S170, M208, the blocking loop between G266 and C270, T301, I314 and K315. Our previous study [18] showed that mutations at S170, M208, I314 and K315 contributed to poor dox control, likely due to increased PLpro stability or activity, and so to obtain true escape scores we must consider the effect of different variant baseline activities in our analysis. Indeed, inspection of the sequence-function map for baseline (leaky) activity showed mutations at S170, M208 and multiple residues at the C-terminus to be impacted by poor dox control (S7 Fig).

To address this issue, we developed a data analysis pipeline that accounts for both leaky expression and the inherent activity of each variant, allowing us to recalibrate drug escape scores accurately. When testing an individual variant by flow cytometry, we measure biosensor cleavage in three conditions: (1) prior to dox induction (leaky expression), (2) after dox induction, and (3) after dox induction in the presence of the inhibitor. To quantify inhibition, we calculate the percentage of cells within the FRET$^-$ gate under each condition and use the following ratio: (FRET$^-_{dox}$ – FRET$^-_{drug}$)/(FRET$^-_{dox}$ – FRET$^-_{leaky}$). In this calculation, a ratio of **1** indicates that the drug is fully inhibitory, while a ratio approaching **0** indicates that the drug is no longer able to inhibit PLpro. We define the **escape score** as the inverse of this ratio, meaning that higher escape scores correspond to greater resistance to inhibition.

The DMS fitness scores define a log-fold enrichment of a variant in the FRET$^-$ gate compared to the FRET$^+$ gate [29]. By setting a reasonable expectation of the proportion of cells containing wildtype PLpro falling in the FRET$^-$ gate before and after dox treatment and with complete inhibition by drug we can convert fitness scores into pseudo-FRET$^-$ proportions for each variant (see Methods and Fig 2A). Along with previously collected activity DMS data [18], we converted fitness scores from leaky expression and inhibitor DMS datasets (in this study) into FRET$^-$ estimates that allowed us to calculate the level of inhibition for each treatment. Since we were interested in those with low inhibition, indicating drug escape, we

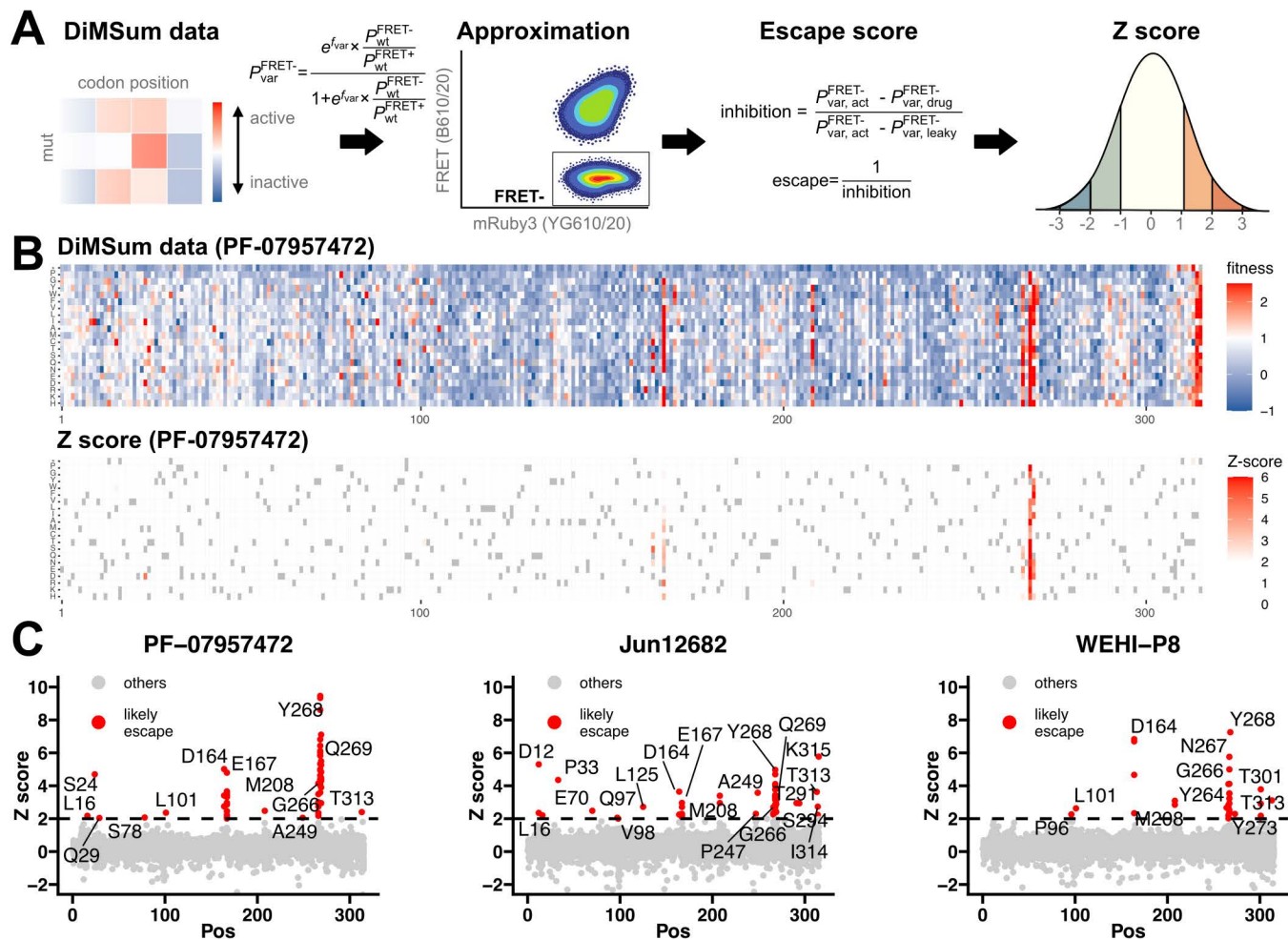

**Fig 2. Data processing pipeline and identification of drug escape hotspots.** (**A**) Transformation of DMS data to FRET- ratio. The raw DiMSum fitness scores are converted to FRET ratios using a custom equation (see methods). An inhibition score is then calculated from the FRET- ratios obtained from activity [18], leaky, and drug treatment DMS data. This inhibition score is inversed to calculate an escape score, which is subsequently normalized into a Z-score, based on the distribution of synonymous wildtype variants. Created in BioRender. Wu, X. (2025) https://BioRender.com/zp95rys (**B**) Comparison of data before (top) and after (bottom) processing. The top panel shows the DiMSum data, while the bottom panel displays data after processing through our pipeline. Variants are plotted by their residue number (x-axis) and mutation (y-axis). Variants with either a DiMSum fitness score over 1 or a Z-score above 2 are highlighted in red. Refer to S4–S6 and S8–S10 Figs for detailed heatmaps. (**C**) Identification of drug escape hotspots. For each drug treatment, the Z-scores (y axis) are plotted for all variants arranged by residue position. Variants with a Z-score greater than 2 are highlighted in red, indicating drug escape variants. Their corresponding residue positions are labeled on the graph. These scores are also plotted in S8–S10 Figs in sequence-function heatmaps.

took the reciprocal of the inhibition score to obtain an escape score. An example of the DMS data before and after this treatment is shown in Fig 2B.

To make scores comparable across datasets with different noise levels, we used synonymous wildtype variant escape scores, which are centered around 1 (indicating no escape). We calculated the standard deviation of these scores and then converted variant escape scores to Z-scores, representing the number of standard deviations from the mean (S1 Dataset - Z-score calculation). The resulting scores are shown in Fig 2C and show positions that are clearly enriched in escape variants. Complete sequence-function maps are shown in the S8–S10 Figs. Using this method, we excluded most

C-terminal mutations as artefacts of leaky expression while identifying hotspots of escape variants for each treatment (Figs 2C and S8–S10). The majority of hotspots were found at residues directly mediating drug binding, with similarities seen in PF-07957472 and Jun12682 drug escape profiles. WEHI-P8 exhibited a distinct escape profile, and although some positions overlapped, the nature of the mutations mediating strong drug escape differed. Mutations at D164 and G266 appeared to impact all three inhibitors, while mutations at M208, N267, and T301 significantly reduced the activity of WEHI-P8. Mutations at E167G, Y268, and Q269 mainly impact PF-07957472 and Jun12682.

### Z-scores measured by DMS accurately reflect PLpro escape from inhibition

To determine how accurately our Z-scores reflect each variant's actual escape from inhibition, we tested selected variants in an orthogonal assay using recombinant protein. We selected a diverse range of PLpro variants for additional testing (listed in Fig 3A). L162R was selected because we expected this residue to be involved in escape due to its proximity to each compound's binding site, however our DMS data suggested it only mildly impacted inhibition. D164C and G266S were selected because they mediated escape from all three compounds. E167G, M208T/A/V/W, Y264F, N267K, Y268R, Y268C, Q269H, C270F, and T301A were selected because they displayed different propensities to mediate escape among each compound. S170F, I314T and K315N were selected because they had higher escape fitness scores in raw data, before correction accounting for leaky expression, and are predicted to not mediate escape by our refined analysis pipeline.

Proteins were expressed in *E.coli* and his-tagged at their N-terminus to enable purification [15]. We confirmed purity, equivalent concentration and size of these proteins by SDS-PAGE analysis and for most variants confirmed the correct mutation was incorporated by native mass spectrometry analysis (S11 and S12 Figs). We assessed compounds with each variant in an assay that measured the ability of PLpro to cleave the substrate Z-RLRGG-AMC, where fluorescence is observed upon cleavage. IC50 values were calculated from dose response curves (S13–S15 Figs) and the average IC50 values across two biological replicates are reported in Fig 3A. The IC50s from wildtype PLpro and each compound were as follows: 7.95 nM (PF-07957472), 39.8 nM (Jun12682), and 10.8 nM (WEHI-P8), all of which align with previously reported literature values [6,19,26]. We compared the IC50 values for all other variants (Fig 3B) and plotted these on a log scale against Z-scores from the DMS analysis (Fig 3C). This resulted in a strong linear correlation, with Pearson coefficients close to or over 0.8, indicating the high predictive value of our DMS analysis pipeline.

The results from our biochemical assays were as expected for the mutations we selected. As predicted, D164C and G266S dramatically reduced the capacity of all three compounds to inhibit PLpro, with ~ 40–400 fold losses in inhibition. L162R, which is near the compound binding site, mildly reduced the activity of all three inhibitors, causing a 4- to 7-fold loss in potency. Since L162R was not identified as a strong escape mutant by DMS, this likely reflects the lower limits of detection for our screen. As predicted by our DMS workflow, E167G, Y268R/C, and Q269H strongly affected PF-07957472 and Jun12682. C270F also mildly affected these two compounds, while not affecting WEHI-P8. M208W, N267K, and T301A selectively reduced the activity of WEHI-P8. Other mutations at M208, such as T/A/V, mildly affected compound activity, with influence skewed towards PF-07957472 and Jun12682. Mutations that were chosen to validate our approach correcting for leaky expression (S170F, I314T and K315N) had no impact on the activity of any inhibitors. Together, these results indicate that our pooled analysis is able to identify and rank compound escape mutations and can measure increases in IC50 of just a few-fold.

### Biochemical properties of variants

Above, we used assays to measure the ability of PLpro to recognize and cleave the RLRGG motif in a linear peptide sequence. However, PLpro also removes post-translational ubiquitin and ISG15 modifications. In both cases, the C-terminal carboxyl group of ubiquitin and ISG15, whose last four residues are LRGG, is covalently linked to the ε-amino group of a lysine residue on the modified protein via an isopeptide bond. Ubiquitin and ISG15 folded domains also participate in substrate

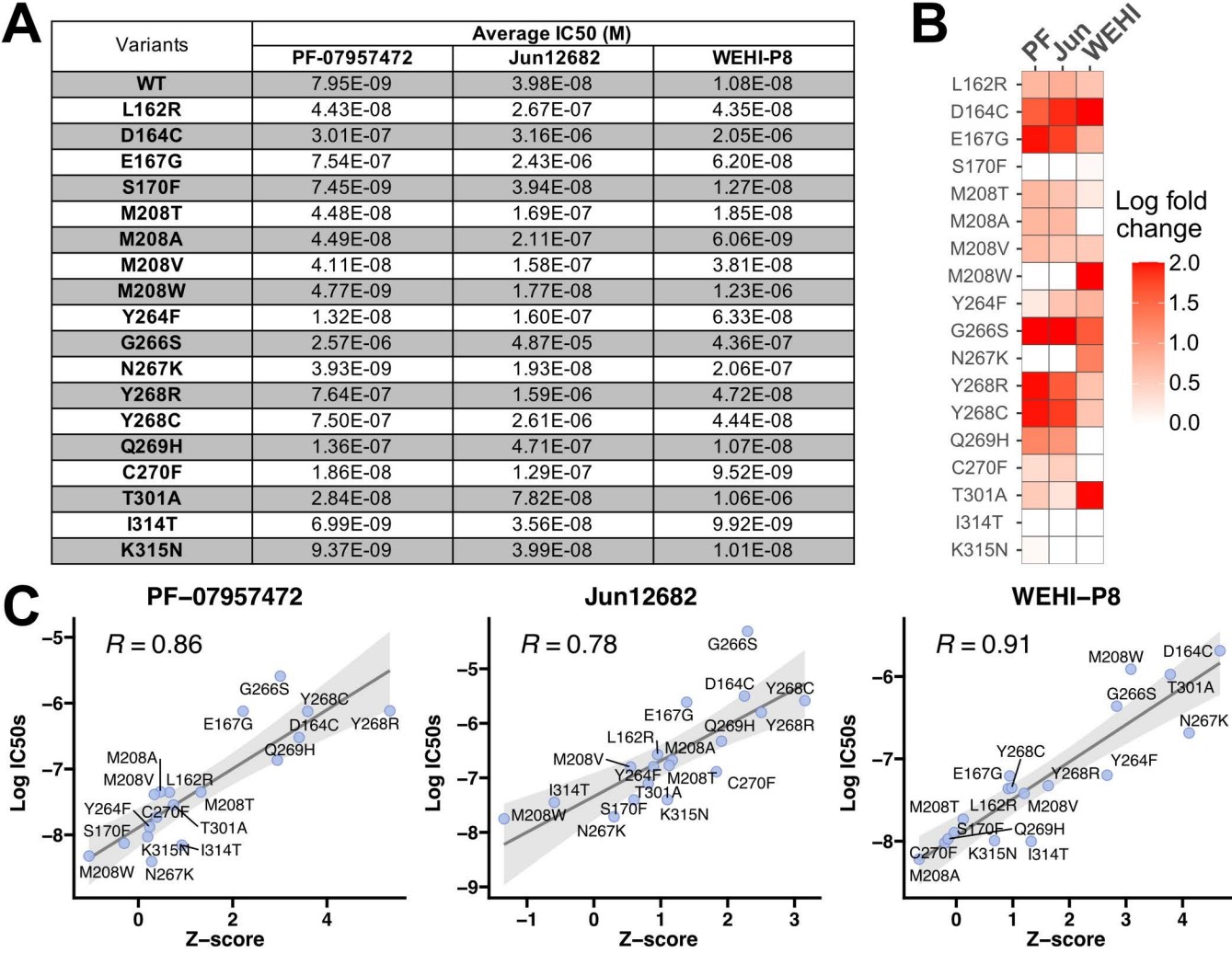

**Fig 3. Inhibitor potency correlates with Z-score.** (**A**) The table displays the IC50 values for 17 variants plus wildtype in response to three compounds PF-07957472 (the 2nd column), Jun12682 (the 3rd column), and WEHI-P8 (the 4th column). The IC50 values represent the mean of two independent biological replicates; corresponding dose-response curves are shown in S13–S15 Figs. (**B**) The heatmap illustrates the log-transformed IC50 fold-change relative to wildtype of the data in a), with a color scale ranging from 0 (white)-2 (red). Rows correspond to different protein variants as labeled, while columns represent the treatment (left-right: PF-07957472, Jun12682, and WEHI-P8). (**C**) The plots show the relationship between Z-scores with log-transformed IC50 values, for each drug treatment (left-right: PF-07957472, Jun12682, and WEHI-P8). The linear regression is performed in R using "lm" function, with the Pearson coefficients displayed in the top left corner of each plot. The shadow area indicates the 95% confidence interval of the regression.

recognition, interacting with PLpro fingers and thumb domains that are distal from the active site [14,30]. Additionally, the DMS activity data reports on cleavage of the cellular biosensor at equilibrium, but not the speed at which equilibrium is achieved. We therefore measured the rates at which PLpro variants cleaved the substrates Z-RLRGG-AMC, Ub-Rhodamine110Gly and ISG15-Rhodamine110Gly over a 3-hour time course to determine if the mutations mediating escape from PF-07957472, Jun12682 and WEHI-P8 impacted catalytic efficiency and/or substrate specificity (S16–S18 Figs).

We used the first 30 minutes of this time course to calculate cleavage efficiency from linear rates (Fig 4A) and summarized the activity as a percentage of wildtype (Fig 4B). All variants retained some catalytic activity, but we observed substantial differences in the rate of substrate cleavage, especially for PLpro variants with mutations that conferred

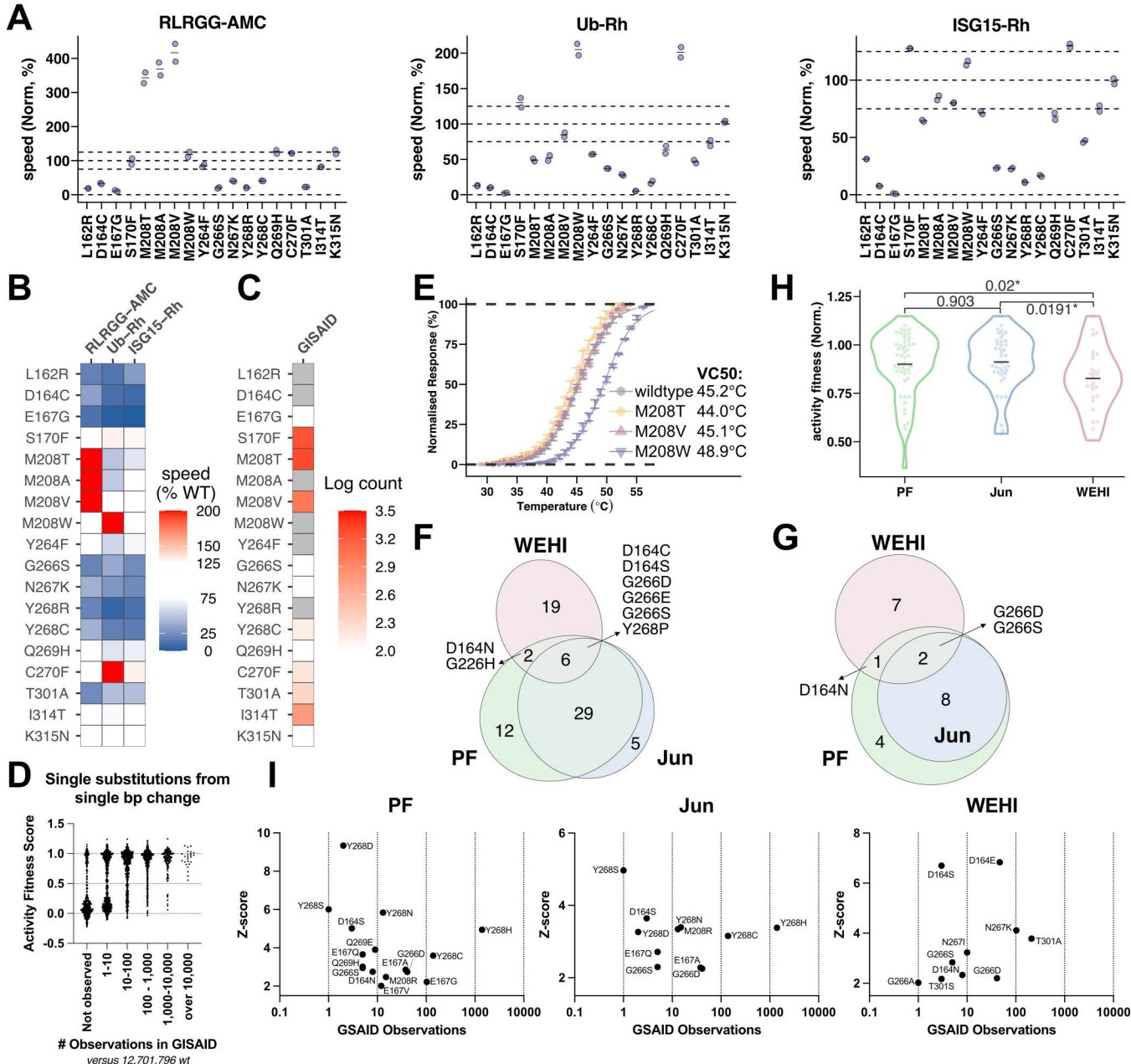

**Fig 4. Biochemical properties of selected variants and inhibitor escape profiles. (A)** Comparison of substrate cleavage speeds of variants with wildtype against different PLpro substrates (as labelled), with data from "no PLpro" control set to 0 and data from wildtype to 100. Two independent biological replicates were measured, with each dot representing the average of three technical replicates. (**B**) Summary of data in A) where speed relative to wildtype is shown on a blue-white-red gradient scale from 0 to 200%. Wildtype-like scores ranging from 75-125% are shown in white. Variants cleaving faster than wildtype are in red, while those cleaving slower are in blue. (**C**) Variants present in GISAID colored according to the number of observations on a log scale from 2.0 (i.e., 100 observations, white) to 3.5 (i.e., ~3200 observations, red). (**D**) Activity of circulating PLpro variants as a function of number of sequences in GISAID dataset. (**E**) Representative thermal shift assay of M208 variants. The data are normalized so that the baseline is 0 and the peak is 100. Three technical replicates are depicted as dots, with error bars indicating mean±SD. The reported VC50 is the average value from two biological replicates. (**F-G**) Area scaled Venn diagrams of drug escape variants from DMS data. F) shows the overlap patterns of confident escape variants identified via DMS data. G) shows variants accessible via a single base-pair mutation. PF-07957472 (green); Jun12682 (blue); WEHI-P8 (red). (**H**) Activity fitness scores from previous work [18] of confident escape variants are plotted by treatment (PF-07957472, green; Jun12682, blue; WEHI-P8, red). The activity scores are normalized so that wildtype has an activity score of 1 and nonsense variants have a score of 0. Violin plots show the distribution of fitness scores of variants within each treatment group, with the short bar indicating the mean value for that group. Pairwise p-values

were calculated using Dunn's test, and p-values below 0.05 are marked with an asterisk to indicate significance. (**I**) Escape variants observed in GISAID dataset with number of observations plotted on the x-axis and Z-score on the y-axis. Vertical lines delineate the same bins shown in (**H**).

escape from the compounds. L162R, D164C, E167G, blocking loop mutants (G266S, N267K, Y268R/C), and T301A slowed PLpro catalytic activity across all three substrates. As in previous work [18], we observed that mutations at M208 differentially affected substrates: M208A/T/V processed the short peptide 3-fold faster than wildtype, while M208W processed ubiquitin~2-fold faster. Similarly, C270F also exhibited faster processing of ubiquitin, and inspection of the leaky expression dataset suggests this variant is mildly leaky in addition to being a mild escape variant for GRL0617-based compounds. Y264F showed poor de-ubiquitinating activity, while the cleavage of other substrates was less impacted, paralleling the effects observed with Q269H. Variants S170F, I314T and K315N that were chosen due to their display of leaky expression rather than as escape variants behaved similarly to wildtype PLpro. Therefore, the variants of interest from the perspective of escape, mediated changes to the ability of PLpro to efficiently cleave substrate.

Whether lowered PLpro activity impacts viral fitness is not determined in our assays. However, millions of viral isolates have been sequenced throughout the course of the pandemic and this data can be mined to determine if any particular mutant has been observed as naturally occurring, and thus unlikely to be catastrophically detrimental to viral fitness. We downloaded the latest GISAID [31] data (July 23rd, 2025) and extracted reads that contained PLpro sequences, ignoring partial reads and those with indels. 16,083,184 sequences passed our filters, with 12,701,796 corresponding to the canonical PLpro sequence upon which our library is based. 3,074,197 contained a single amino acid (aa) change, 292,924 had double aa substitutions, 13,604 had triple aa substitutions and only 682 sequences contained more than three mutations. Among the 17 protein variants we tested in isolation, E167G, S170F, M208T, M208V, G266S, N267K, Y268C, Q269H, C270F, T301A, I314T and K315N have been observed in patient isolates (Fig 4C, S1 Dataset). S170F, M208T and M208V, while still rare in the context of the sequences reported, were more frequently observed compared to the other variants. E167G, the variant that had the lowest activity against all three substrates, had been observed 103 times. While activity over 30 minutes in biochemical assays is very slow for this variant, in cells it is able to cleave a significant proportion of biosensor, with a fitness score of 0.59, where 1 is active and 0 is inactive (S19 Fig) [18]. The observation that this variant is present in circulating virus, albeit with reduced catalytic efficiency, indicates that even slow proteolysis may be able to support viral replication. The lowest activity variant as measured by our cellular assay that supports escape from PF-07957472 was G266Y with a fitness score of 0.365 and an associated error of 0.077.

To determine if low activity variants are permissible in the context of viral fitness, we stratified the number of observations from the GISAID dataset and plotted fitness scores for variants that are accessible by a single base pair change that were: absent from the dataset, observed 1–10 times, 10–100 times and so forth, to visualize the spread of activities at different observation rates (Fig 4D). We saw strong enrichment of functionally impaired variants in the bin containing variants that were absent from the GISAID dataset, indicating that most single base pair substituted variants that support function have been observed. Sequencing errors may account for many observations; however, we saw strong enrichment for active variants starting in the gate with 10–100 observations. Poorly active variants were found even in the 1,000–10,000 observation bin but were absent at higher counts. While it is difficult to delineate an absolute activity threshold required for viral fitness from these data, it is likely that poor proteolytic activity can be overcome by the virus in the context of the extended timeframes of natural infection coupled with the proximity of covalently linked substrates during polyprotein cleavage. When the virus is under selective pressure, second and even third site mutations have been observed that likely act to restore activity to variants that escape selection pressure at the cost of fitness [32]. Together these data provide a framework for discerning variants that drive drug escape from those that stabilize or enhance enzyme structure and function.

The higher frequency of M208T/V observed in circulating variants is interesting in the context of our observation that this mutation increases the rate of Nsp cleavage, while mildly impacting ubiquitin and ISG15 cleavage. We previously

observed that M208W improved thermal stability, while M208A/S stability mimicked wildtype PLpro [18]. We compared the stability of M208T/V with M208W and wildtype PLpro and found that while M208W denatured 3.7°C above wildtype PLpro, M208T/V, like M208A/S in our previous work [18], did not show altered thermostability (Fig 4E).

**Comparison of inhibitor escape variants**

Having validated our DMS pipeline with selected variants, we considered the dataset as a whole. We selected variants with a Z-score higher than 2 for each compound. Some of these variants had low counts and high errors, so we filtered these variants from the analysis (S20 Fig and S1 Dataset - Confident escape variants). The filtered variants are marked with an X in S8–S10 Figs.

To visualize the intersection of escape variants from each compound, we used an area-proportional Venn diagram where the size of the ellipse represents the number of variants and the area of overlap approximates the number of escape variants that are shared among compounds (Fig 4F and S1 Table). The diagram revealed a near-complete over-lap between PF-07957472 and Jun12682, while WEHI-P8 was affected by a largely unique set of escape variants. The escape variants that affected all three compounds were primarily found at D164 and G266 (Fig 4F).

We next evaluated only escape variants that could arise from a single nucleotide change, making them more likely to occur in circulating variants once selective pressure is applied. We used the original Wuhan strain sequence as a base (Genebank access code: NC_045512.2) [33] and identified escape variants accessible with a single base-pair change (S21 Fig). Selecting only high-confidence escape variants accessible by a single base-pair change, we see that only G266D and G266S can escape all three compounds, with most of the remainder being unique to WEHI-P8 or affecting both PF-07957472 and Jun12682 (Fig 4G and S1 Table).

In similar work on Mpro, escape mutants were identified for nirmatrelvir and ensitrelvir [34]. An interesting observation was that escape mutants generally had lower-than-expected functional scores. This suggests that, in order to successfully evade a drug while maintaining viral replication, Mpro must accumulate both an escape mutation and a compensatory sta-bilizing mutation(s). We compared the functional scores [18] of escape variants to PF-07957472, Jun12682, and WEHI-P8 (Fig 4H) and found that, on average, mutations that prevented WEHI-P8 activity had lower functional scores than those affecting PF-07957472 and Jun12682. This suggests that evasion of WEHI-P8 may require more compensatory changes to PLpro than evasion of the other two compounds.

Finally, we plotted the Z-score for each escape variant against the number of observations in GISAID data to gauge the likelihood of resistance emerging from each compound (Fig 4I). Results showed every escape variant that was accessible by a single base pair mutation (except Y264F) had been seen at least once. No escape variant that required two or more base pair changes was observed except D164S, which can escape all three compounds and has been observed 3 times. We provide a table of all predicted escape variants, their Z-score, activity fitness scores, accessibility with only one base pair change and the number of times they were observed in GISAID data (S2 Table).

## Discussion

Despite the first small-molecule therapeutics targeting SARS-CoV-2 being launched in record time, drug-development efforts continue, both to broaden our therapeutic arsenal and to prepare for future pandemics [35,36]. Here we have assayed the landscape of PLpro single-site substitutions that can mediate escape from two GRL0617-based compounds and a recently reported novel scaffold exemplified by WEHI-P8 [26]. In previous work [18], we also assessed escape variants from the 3k/5c scaffold [15,17]. While we were able to confidently identify positions where multiple variants medi-ated escape, we were less confident in the ranking among identified escape variants than we are for the higher potency compounds assessed here. Together these datasets identify the residues critical to emerging PLpro-targeting lead com-pounds, and in the case of compounds reported here, assess how different aspects of the structure activity relationship affect efficacy.

GRL0617-based, 3k/5c and WEHI-P series compounds all bind within the S4 pocket (Figs 1A and S2), while making unique contacts with the surrounding residues. Broadly speaking, we have identified the following 10 residues as critical hotspots among GRL0617-based, 3k/5c and WEHI-P8 compounds: D164, E167, M208, P247, Y264, G266, N267, Y268, Q269 and T301 (S8–S10 and S22 and 2C Figs). When viewed together on the PLpro structure, these residues surround all sides of the S4 pocket, form a substantial part of the blocking loop and make critical contacts with each compound as judged from published crystal structures [6,15,19,26,37]. Despite overlapping binding sites, when all the escape mutants are collated only 6 out of 74 confidently identified variants are capable of escaping all three compounds (Fig 4F). These mutations (D164C/S, G266D/E/S, & Y268P) occur at 3 positions and our previous data [18] suggests that 3k/5c is likely to be impacted by D164C/S & Y268P, but not G266 mutations (S22 Fig).

## The blocking loop and variants at G266, N267, Y268, Q269

Four out of the ten positions with distinct impacts on each compound are found in the blocking loop of PLpro. The blocking loop is an interesting structural feature that is flanked by two glycines; G266 likely affords the entire blocking loop flexibility and G271 lines the narrow channel from S4 to the active site, such that mutations in this position would block substrate access to the active site. Most mutations at these glycines impact PLpro activity (S19 Fig) but the intervening residues forming the blocking loop are not conserved among coronaviruses and can be mutated without impacting proteolytic activity [18,38,39] (S19 Fig). In crystal structures, the conformation of the blocking loop is variable depending on crystal contacts and engagement of ligands in the S4 pocket. Both GRL0617-based and 3k/5c compounds engage the blocking loop similarly, flipping the loop into a closed position where the sidechain of Y268 T-stacks against the naphthyl ring of both compound series. The substituted phenyl ring of both GRL0617-based compounds in this conformation aligns with the Q269 sidechain to maintain Van der Waals packing (Fig 1A), while the 3k/5c phenyl group sits over the Q269 backbone flipping the sidechain out towards solvent (S2 Fig). In contrast, the blocking loop adopts a different conformation when binding WEHI-P series compounds (as exemplified by the WEHI-P4 crystal structure [26]) with respect to the placement of the Y268 sidechain as well as displacement of the blocking loop backbone from residues N267-C270 (Fig 1A). In this conformation, the sidechain of N267 creates bidentate hydrogen bonds to the backbone nitrogen atoms of Q269 and C270, stabilizing the altered blocking loop conformation and pointing the Q269 sidechain out into solvent.

Given the different structural features of the blocking loop when engaging GRL0617-based and WEHI-P series compounds, we can see explanations for the distinct susceptibility of each series to different mutations. GRL0617-based compounds (and from previous work 3k/5c) are affected by almost all mutations at Y268, while WEHI-P8 is only strongly impacted by the single variant Y268P. While mutations at Q269 interfere with GRL-0617-based compounds, likely through steric hindrance or charge repulsion mechanisms, WEHI-P8 tolerates these changes, and is instead uniquely sensitive to N267 alterations. We interpret Y268P and N267X variants as impacting the backbone conformation of the blocking loop that is uniquely induced by WEHI-P series compounds.

## Key interactions with D164, Y264 and the backbone of Y268

Two backbone nitrogens of the peptidic substrate ($Leu_{P4}Arg_{P3}Gly_{P2}Gly_{P1}\downarrow$) form critical hydrogen bonds within the S4 pocket. The backbone nitrogen of $Leu_{P4}$ hydrogen bonds with the side chain of D164 and the backbone nitrogen of $Arg_{P3}$ engages both the hydroxyl of Y264 that sits at the base of the S4 pocket and the backbone carbonyl of Y268 found on the blocking loop. GRL0617-based, 3k/5c and WEHI-P8 compounds all maintain interactions with these residues: GRL0617 through the amide group that links the two ring systems and WEHI-P8 through the tertiary amine of the piperidine ring and the hydroxyl of the cyclohexanol. The lower potency 3k structure shows a clear hydrogen bond between nitrogen of the amide group to the carbonyl of Y268 (S2 Fig), with the piperidine nitrogen close to D164 and only Van der Waals contacts with the sidechain of Y264. Given these core interactions that are common among substrate and small molecule interactions, it is not surprising that D164 variants impact all compounds. While most variants at this position ablate PLpro

activity, those that maintain hydrogen bonding potential are tolerated. Of these, only D164E and D164N are accessible by a single base-pair mutation. D164E strongly inhibited WEHI-P8 but not GRL0617-based compounds, and D164N is one of the few mutations to which all inhibitors are sensitive.

## Unique interactions with E167, M208, P247, and T301

The structure-activity relationship of GRL0617-based compounds illustrates several approaches that have been undertaken to optimize potency. A basic amine substituent is often added to the phenyl ring of GRL0617 to interact with the sidechain of E167 [5,6,19,24], bulky substituents have been added to optimize occupancy of the S4 site [5,19], and modification of the naphthyl into multi-ring systems has been performed to extend into a groove formed by the blocking loop [5,6]. Combinations of these strategies have led to the discovery of various potent PLpro compounds, such as HL-21 [25] & GZNL-36 [5] in addition to the compounds studied here. Jun12682 additionally includes a pyrazole ring substitution intended to occupy the same space as ubiquitin residue Val70 as observed in the co-crystal structure [6,14]. WEHI-P8 extends out from the S4 pocket towards M208, altering the M208 and R166 sidechain rotamers to enlarge the S4 pocket and enable binding [26].

In the positions found outside the blocking loop and at key substrate interaction residues, we see how PLpro may escape some of these strategies. PLpro tolerates variants at E167, with all but E167P retaining some level of enzymatic activity (S19 Fig). The escape data shows that PF-07957472 and Jun12682 are both dependent on the basic amine substituents added to the GRL0617 scaffold to improve potency. M208 is also tolerant of mutations (S19 Fig) and circulating variants have been observed with M208T and M208V mutations (Fig 4C), which increase the speed of proteolysis for peptide substrates while mildly slowing ubiquitin and ISG15 removal activity (Fig 4B). M208W conversely promotes increased PLpro stability and improves deubiquitinating activity but has not been observed in circulating variants. WEHI-P8 inhibition can be evaded by variants with aromatic substitutions at M208, which instead enhance binding of GRL0617-based compounds. In contrast, GRL0617-based compounds showed mild escape with non-aromatic substitutions at M208, resulting in mutually exclusive escape profiles at this position for the tested inhibitors.

P247 and P248 lie at the rim of the S4 pocket, distal from the active site, with the aromatic moieties of GRL0617-based, 3k/5c and WEHI-P series compounds positioned nearby. WEHI-P8 forms the closest contacts with P247. Despite this, we only saw mild Jun12682 escape at P247, suggesting the compounds are fairly tolerant to mutation at this position. WEHI-P8 was instead uniquely sensitive to mutations at T301. T301 is a conserved position with only V/A/C and S variants retaining activity (S19 Fig). All these mutations ablate WEHI-P8 activity (S19 Fig), despite no direct contacts predicted from the WEHI-P4 PLpro crystal structure. Instead, T301 forms a hydrogen bond to the backbone carbonyl and nitrogen of A246, which likely stabilizes the turn of P247 and P248 to provide an appropriate platform for the naphthyl ring of WEHI-P8 to bind.

## The relationship between drug escape and viral fitness

A study published while this work was under review has directly examined the question of whether PL$^{pro}$ mutations that confer drug resistance are fully compatible with viral replication and therefore likely to gain traction in widespread circulation. Using a structure-guided, candidate-based approach to select and test naturally occurring variants likely to escape GRL0617-based inhibitors, Tan and Zhang et al also identified E167, Y268 and Q269 as primary drug-escape hotspots [32]. They further showed by serial passage of SARS-CoV-2 virus in the presence of increasing amounts of Jun12682 that E167G and Q289H mutants emerged spontaneously and were equally cross-resistant to PF-07957472. This result underscores the significance of our data showing that WEHI-P8 retains significant activity against E167G and full activity against Q269H. These mutant viruses exhibited mildly attenuated replication rates, consistent with the observation in their study and in ours that the enzymatic activities of E167G and Q269H mutant PLpro recombinant proteins are slightly reduced. Interestingly, second-site mutations at V98, L152 and T168 also emerged from the serial passage experiments

that significantly enhanced viral fitness in the presence of Jun12682 even though these mutations do not independently confer detectable drug resistance, which our data also confirm. This highlights the reality that even mild fitness costs associated with drug resistance can select for compensatory mutations mitigating those effects, and it suggests that further DMS screening of second-site mutations on the background of the strongest drug resistance mutations may also have significant predictive value. Finally, as we reported in a previous study [18] and extended in the experiments described here, Tan and Zhang et al also identified that some M208 variants confer increased enzymatic activity and mildly affected inhibition by GRL0617-based compounds. The strong agreement in results between their candidate-based approach and our unbiased screening approach makes these two studies highly complementary and bolsters confidence in the real-world relevance of the unique WEHI-P8 resistance profile.

Together these data exemplify the shared and distinct ways some of the most potent small-molecule inhibitors bind PLpro. All engage D164, which is also crucial for the engagement of the natural substrate. The three compounds tested here are also critically dependent on G266, which does not contact these compounds but instead highlights the importance of the correct positioning of the blocking loop to lock in binding. The sidechain of Y268 is particularly important for GRL0617-based and 3k/5c compounds. In contrast, WEHI-P8 is not dependent on direct side-chain interactions with residues on the blocking loop, but variants at G266 and N267 suggest that the backbone of the blocking loop does need to be appropriately positioned to engage the hydrogen bond with the cyclohexanol oxygen. While each compound series exhibits distinct structural vulnerabilities that could potentially be exploited by SARS-CoV-2 to develop resistance, the diversity in binding modes suggests that if one inhibitor loses efficacy, others with alternative interaction mechanisms may still retain activity. Therefore, developing diverse chemical scaffolds in the context of PLpro drug-discovery provides a strategic advantage by reducing the likelihood of cross-resistance in the clinic.

## Materials and methods

### Vectors

The vectors used for FRET biosensor, DMS assay, and recombinant PLpro expression have been described in our previous work [18]. In brief, we first synthesized the FRET biosensor (mCover3-TLKGGAPTKV-mRuby3) as a gene block (IDT) and subsequently inserted it to the retroviral expression vector (pMX-Gateway-IRES-Hygro, a gift from Andrew Brooks at the University of Queensland) using LR cloning (Invitrogen, #11791020).

We expressed PLpro in the pOPINB expression vector, which was a gift from Ray Owens (Addgene plasmid #41142). We introduced site-mutations into this expression vector using the NEB Q5-SDM kit (NEB, #E0554S) following the manufacturer's instructions.

The FU-tetO-PLpro-rtTA-2A-Puro vector [18] was used for the expression of PLpro and its variants, including those in the DMS libraries. Approximately one third of the DMS library was directly synthesized by Twist Bioscience. The remaining variants were generated using our modified MiTE mutagenesis [40] protocol which involves the integration of degenerate oligo pools (IDT) into a backbone through HiFi assembly. Each variant in the library was uniquely tagged with a 16 bp barcode, and the barcode-variant association was determined via PacBio sequencing [18].

### Tissue culture

**Cell line.** Verified HEK293T cells (Cellbank Australia, #12022001) were cultured in DMEM (GIBCO, #10313039) supplemented with 2 mM glutamine (GIBCO, #25030081) and 10% FBS (GIBCO, #2526728RP).

**Retrovirus production and transduction.** HEK293T cells were transfected with transfer vector and retroviral packaging vectors MMLV-gag-pol and VSVg or lentiviral packaging vectors RSV-REV, pMdi and VSVg (all packaging vectors were gifts from Marco Herold, WEHI), using Calcium Phosphate precipitation. 2 days after transfection, the supernatant was collected and filtered through a 0.45 μm filter, aliquoted, and stored at -80˚C for future usage.

On the day of transduction, thawed virus was diluted in complete DMEM containing polybrene (Sigma Aldrich, #TR-1003-G) at a final concentration of 4 µg/ml and added to HEK293T cells at approximately 50% confluency. Cells were spun at 37°C for 45 minutes at 1000 *g* to encourage transduction before being incubated overnight at 37°C. The next day, the medium was replaced, and cells were evaluated two days after transduction.

To establish the biosensor cell line, we transduced the retrovirally packaged FRET biosensor into HEK293T cells, before treating them with 180 ng/µl Hygromycin (Merck, US1400052) for 2 days to eliminate non-transfected cells. Cells were subsequently sorted twice based on a medium FRET signal. To establish the cell line to measure PLpro inhibition (PLpro-biosensor cells), we transduced the biosensor cell line with wildtype PLpro in the expression vector of FU-tetO-PLpro-rtTA-2A-Puro at an MOI of ~ 0.1 to minimize multiple integrations. Cells were selected with 2 ng/µl Puromycin (Thermo, #A1113803) for 2 days to remove non-transduced cells. Finally, these cells were aliquoted and stored in liquid nitrogen in a cryoprotective medium composing of 90% FBS and 10% DMSO.

**Cellular inhibitor assay.** For each assay, a fresh batch of PLpro-biosensor cells were thawed. Compounds, such as Jun12682 [6] (MedChem Express, #HY-157403), PF-07957472 [19] and WEHI-P8 [26] (sourced in-house), were serially diluted in DMSO to create an 8-point, 3-fold dilution series starting at 2 mM (to achieve a final concentration of 20 µM in the cellular assay). For each dilution, 1.5 µl was added to three separate wells in 96-well, flat-bottom, tissue-culture-treated plates. In addition, 1.5 µl DMSO was added to designated wells as controls. Recovered cells were then harvested at ~ 80% confluence and diluted to 1.67 x 10^5 cell/ml. A volume of 150 µl of the cell suspension (~ 25,000 cells) was added to the DMSO control wells as no doxycycline control; dox (Sigma Aldrich, #D5207) was added to the remaining cells to a final concentration of 300 ng/ml to induce PLpro expression and 150 µl of cell mixture was seeded to all other wells. After overnight incubation at 37°C with 10% CO2, the cells were detached and analyzed by flow cytometry at the WEHI FACS facility to determine the FRET+ percentage of cells at each inhibitor concentration.

We normalized the FRET+ percentage so that the value for no inhibitor treatment was 0 as negative control, and the value for no PLpro expression (no dox treatment) was 100 as positive control.

$$k = \frac{100}{\left(\text{FRET}^+_{\text{no-PLpro}} - \text{FRET}^+_{\text{no-inhibitor}}\right)}; \; b = -1 * k * \text{FRET}^+_{\text{no-inhibitor}}$$

(1)

$$\text{Inhibition} = k * \text{FRET}^+ + b$$

(2)

Next, the inhibitor concentrations (in molar units) were transformed to their logarithmic values. Using these log-transformed concentrations along with the corresponding inhibition percentage, a dose-response curve was fitted in R studio v4.4.2, using the nlsLM function from minpack.lm package. The formula is shown below:

$$\text{Inhibition} = \frac{100}{\left(1 + 10^{\left(\text{EC50}_{\text{log}} - \text{drug-concentration}_{\text{log}}\right)}\right)}$$

(3)

The EC50 for each compound was then determined using:

$$\text{EC50} = 10^{\text{EC50}_{\text{log}}}$$

(4)

## DMS workflow

In a manner similar to our previous PLpro DMS screen [18], the PLpro library was first mixed with an unrelated vector (pFGH1-UTG-mTagBFP2) at a 1:2 ratio to avoid barcode swapping [41]. This mixture was then transduced into the bio-sensor cell line at an MOI of ~ 0.2 to minimize multiple integration. There were around 10 cells representing each variant.

2 days post-transduction, we treated cells with Puromycin (Thermo, #A1113803) at a final concentration of 2 ng/μl for an additional two days. Subsequently, live cells were harvested and split into four equal portions to receive the following treatment: blank (no compound), 300 ng/ml dox with 3 μM PF-07957472, 3 μM WEHI-P8, or 10 μM Jun12682. The next day, cells were harvested and sorted into FRET+ and FRET- gates for RNA extraction – except those receiving no treatment (blank), which were re-seeded with 300 ng/ml dox for 4 h to induce PLpro mRNA expression before RNA extraction.

RNA extraction was performed using ReliaPrep RNA Miniprep Systems (Promega, #Z6012) following the manufacturer's instructions. The isolated RNA was reverse transcribed using a primer that annealed 3' to the barcode; this primer also incorporated a unique molecule identifier (UMI) and an overhang adapter for index primer annealing. After reverse transcription, cDNA was amplified with a second overhang primer (annealing the 5' side of the barcode and 3' to the PLpro stop codon) along with a reverse overhang primer. The PCR products were than indexed in triplicate using the flanking overhangs to allow multiplexed sequencing. Finally, the indexed samples were sent for single-end sequencing using the NextSeq 3000 system (Illumina) at the WEHI Genomics Hub, achieving an average of approximately 60 reads per barcode for each sample.

## Illumina data processing and scoring

Fastq files were provided by the WEHI Genomics Hub. We first demultiplexed and trimmed these files using cutadapt v4.9 [42], retaining only reads containing both an intact 16 bp barcode and 16 bp UMI. Next, we deduplicated the reads based on their UMIs using UMI-tools [43], using the cluster method and an edit-distance-threshold of 1. The deduplicated files were used for calculating fitness scores by comparing variant abundance between FRET+ and FRET- populations, using DiMSum [29], in conjunction with long read-derived barcode lookup tables to map barcodes into variants [18]. In DiMSum, the quality filter was set at 15 and no minimum read number was set. Technical replicates were combined after confirming concordance among samples with triplicate indexes. Experimental replicates were defined as selections from independent transductions.

To allow for easier interpretation of the data, we scaled fitness scores and their associated errors so that the score for wildtype PLpro was 1 ($F_{WT}$) and the mean fitness score of the population of nonsense variants was 0 ($F_{avg-nonsense(res1-305)}$) using the formula below:

$$k = \frac{1}{F_{WT} - F_{avg-nonsense(resi1-305)}}; \; b = -1 * k * F_{avg-nonsense(resi1-305)} \tag{5}$$

$$\text{fitness} = k * F + b; \; \text{error} = k * \text{sigma} \tag{6}$$

The normalized data was visualized as sequence-function maps in R studio v4.3.3, using ggplot2.

## Back calculation of FRET- proportions from fitness data

DiMSum scores variants based on the read counts for each variant in the FRET- and FRET+ flow cytometry gates using the following equation (29):

$$f_{var} = \ln\left(\frac{N_{var}^{FRET-}}{N_{var}^{FRET+}} \times \frac{N_{wt}^{FRET+}}{N_{wt}^{FRET-}}\right) \tag{7}$$

Where $f_{var}$ is variant fitness, $N_{var}^{FRET-}$ is the variant read counts in the FRET- gate, and $N_{var}^{FRET+}$ is the variant read counts in the FRET+ gate. $\frac{N_{wt}^{FRET+}}{N_{wt}^{FRET-}}$ normalizes each dataset so that wildtype returns a score of 0.

From this equation we can extract $N_{var}^{FRET-}$ if we assume:

The ratio of $N_{var}^{FRET-}$ over $N_{var}^{FRET+}$ approximates the ratio of cells in the FRET- ($P_{var}^{FRET-}$) over the FRET+ gate ($P_{var}^{FRET+}$), and that $P_{var}^{FRET-} + P_{var}^{FRET+} = 1$ because all cells would be found in one or the other gate.

This assumption allows the fitness equation to be rearranged to the following.

$$P_{var}^{FRET-} = \frac{e^{f_{var}} \times \frac{P_{wt}^{FRET-}}{P_{wt}^{FRET+}}}{1 + e^{f_{var}} \times \frac{P_{wt}^{FRET-}}{P_{wt}^{FRET+}}}$$

(8)

where $P_{wt}^{FRET-}$ / $P_{wt}^{FRET+}$ can be estimated for each dataset based on prior knowledge of how wildtype PLpro behaves in our cellular assays. We estimate that in our datasets, **95% of cells** fall within the FRET⁻ gate when wild-type PLpro is induced with doxycycline, **10% of cells** do so in the absence of doxycycline due to leaky expression, and **3% of cells** fall within the gate under complete inhibition.

After determining the $P_{var}^{FRET-}$, for Activity, Leaky and Drug Inhibition datasets we calculate inhibition as follows:

$$inhibition = \frac{P_{var,\ act}^{FRET-} - P_{var,\ drug}^{FRET-}}{P_{var,\ act}^{FRET-} - P_{var,\ leaky}^{FRET-}}$$

(9)

The escape score was then calculated as the reciprocal of the inhibition level.

$$escape = \frac{1}{inhibition}$$

(10)

Finally, we standardize the data by calculating a Z-score based on the distribution of the synonymous wildtype population:

$$Z_{score} = \frac{escape_{var} - escape_{mean-syn}}{sd_{syn}}$$

(11)

## Recombinant protein purification

PLpro constructs in the pOPIN-B backbone were transformed into BL21/DE3 (NEB, #C2527I) and cultured in Super Broth (WEHI Bioservices) supplemented with 50 µg/ml Kanamycin (Gibco, #11815032) at 37˚C until an optical density of 0.8 was reached. The culture temperature was then lowered to 18˚C while shaking. Approximately 1 hour later, expression was induced by adding 300 µM IPTG (Bioline, #BIO-37036) and allowing the culture to express PLpro for 16–18 h.

Cells were harvested by centrifugation and resuspended in lysis buffer containing 50 mM Tris·HCl (Invitrogen, #15506017) at pH 7.5, 500 mM NaCl (Supelco, #1.93206), 10 mM Imidazole (Sigma Aldrich, #I202) at pH 8.0, 5 mM beta-mercaptoethanol (Merck, #M6250-100ML), and freshly-made lysozyme (Sigma Aldrich, #L6876). The cell suspension was sonicated and then centrifuged at 40,000 *g*, 4˚C for 30 minutes to clear the lysate. The resulting supernatant was applied to a gravity column packed with His-Tag Purific Resin (Roche, #05893682001). After washing, the bound protein was eluted with a buffer comprising 50 mM Tris·HCl (pH 7.5), 500 mM NaCl, 300 mM Imidazole (pH 8.0), and 5 mM beta-mercaptoethanol. The eluate was desalted using a PD-10 column (Cytiva #17-0851-01) into storage buffer (20 mM Tris·HCl pH7.5, 50 mM NaCl, 1 mM TCEP (Merck, #646547-10X1ML)). Protein purity was assessed by loading ~3 µg of protein onto a 4–12% NuPAGE protein gel (Invitrogen, #NP0322BOX) at 180 V for 35 minutes, using SeeBlue Plus2 (Invitrogen, #LC5925) as the molecular weight marker. Finally, the concentrated protein was aliquoted, flash-frozen and stored at -80°C.

## Native mass spectrometry

Native MS was conducted at the WEHI Proteomics Facility. In brief, proteins were separated using reverse-phase chromatography on a 25 cm ProSwift RP-4H monolith column (Thermo Scientific) with a micro-flow HPLC (Ultimate 3000). The HPLC was coupled to a Maxis II Q-TOF mass spectrometer (Bruker), which was equipped with an electrospray ionization

(ESI) source. Proteins were directly loaded onto the column for an online buffer exchange with buffer A (99.9% Milli-Q water, 0.1% formic acid [FA]), followed by elution with a linear gradient from 2 to 90% buffer B (99.9% acetonitrile (ACN), 0.1% FA). Data analysis was performed using DataAnalysis version 5.2 (Bruker), and proteins were deconvoluted using the maximum entropy method.

## Recombinant PLpro inhibition assay

The PLpro substrate Z-RLRGG-AMC acetate (Sigma Aldrich, #SML2966) was used to evaluate inhibitor dose response. Inhibitors were initially pre-diluted in DMSO at 50x their final concentrations and tested in a 10-point series with 3-fold dilutions, starting at a top final concentration of 2 µM. 120 nL of inhibitor was first spotted into wells (Echo acoustic dispenser, LabCyte) of a 384-well black plate (Corning #3820), with the help of the WEHI National Drug Discovery Centre (NDDC). We then added PLpro and Z-RLRGG-AMC to initiate the reaction (6 µl of 10 nM PLpro and 300 nM Z-RLRGG-AMC in 50 mM HEPES (Gibco, #11344041, pH 7.5), 0.1 mg/ml bovine serum albumin (Sigma Aldrich, #A7030), 150 mM NaCl, 2.5 mM dithiothreitol (Invitrogen, #15508013)). We incubated the plate at room temperature in the dark for 4 h and recorded the AMC fluorescence on a ClarioStar Plus (BMG Labtech, 380 nm excitation; 445 nm emission). We normalized the emissions so the top was 100 indicating no inhibition and bottom was 0 indicating no PLpro:

$$k = \frac{100}{\text{Fluoro}_{no-inhibition} - \text{Fluoro}_{no-PLpro}}; \quad b = -1 * k * \text{Fluoro}_{no-PLpro} \tag{12}$$

$$\text{AMC.Emi}_{Norm.} = k * \text{Fluoro} + b \tag{13}$$

Dose-response curves were fitted in R studio v4.4.2 using the nlsLM function from the minpack.lm package. The formula was:

$$\text{AMC.Emi}_{Norm.} = \frac{100}{\left(1 + 10^{\left(\text{drug-concentration}_{log} - \text{IC50}_{log}\right)}\right)} \tag{14}$$

The IC50 value was subsequently calculated using the formula:

$$\text{IC50} = 10^{\text{IC50}_{log}} \tag{15}$$

## PLpro substrate cleavage assays

We performed cleavage assays in a final volume of 6 µl per wells on a 384-well black plate. PLpro was incubated with its substrates, and fluorescence was recorded every 5 minutes over 3 h on a ClarioStar Plus (BMG Labtech). For AMC fluorescence, measurements were made with an excitation wavelength of 380 nm and an emission wavelength of 445 nm. For Rhodamine110Gly fluorescence, excitation was set at 487 nm and emission at 535 nm.

For assays with R-LRGG-AMC, the buffer was composed of 50 mM HEPES (pH 7.5), 0.1 mg/ml bovine serum albumin, 150 mM NaCl, and 2.5 mM dithiothreitol. For assays with Ub-Rh110Gly and ISG15-Rh100Gly, the buffer was composed of 20 mM Tris (pH 8.0), 0.03% BSA, 0.01% Triton X-100 (Merck, #X100-6X500ML), and 1 mM L-glutathione (Sigma, #G4251).

The following concentrations used in each assay were: 10 nM PLpro was incubated with 300 nM R-LRGG-AMC; 5 nM PLpro was with 62.5 nM Ubiquitin-Rhodamine110Gly (UbiQ, UbiQ-002); 0.25 nM PLpro was with 33.5 nM ISG15-Rhodamine110Gly (UbiQ, UbiQ-127).

Fluorescence data were plotted over time in R studio v4.4.2 using the ggplot2 package. To calculate the initial velocity, we calculated the slope of the first 30 minutes of each fluorescence curve using the lm function in R studio v4.4.2. We normalized the slope information for ease of comparison, so that the background signal (no PLpro) is 0 and wildtype activity is 100:

$$k = \frac{100}{\left(\text{Slope}_{\text{WT}} - \text{Slope}_{\text{no-PLpro}}\right)}; \ b = -1 * k * \text{Slope}_{\text{no-PLpro}} \tag{16}$$

$$\text{Speed}_{\text{Cleavage}} = k * \text{Slope} + b \tag{17}$$

## Thermal shift assay

We diluted proteins in buffers (20 mM Tris·HCl (pH 7.5), 50 mM NaCl, 1 mM TCEP) to a final concentration of 10 µM. Each protein was then mixed with SYPRO orange (Supelco, #S5692) from a 250x stock to achieve a final dye concentration of 10x, using a 24:1 dilution (protein:dye). The mixture was centrifuged at full speed for 10 minutes and 25 µl of the supernatant was transferred in triplicate into PCR tubes (GeneBio Systems, #PCR-4W). The thermal shift assay was performed on a Corbett Real Time PCR machine. The temperature was increased from 25°C to 95°C at a rate of 1°C per minute, and fluorescence readings were taken at each 1°C. Subsequently, the fluorescence data was normalized so that the maximum signal corresponded to 100 and the minimum to 0. These normalized values were then used for nonlinear regression curve fitting in R studio v4.4.2 using the nlsLM function from the minpack.lm package with the formula below:

$$\text{Response}_{\text{Norm.}} = \frac{100}{\left(1 + e^{\frac{\text{VC}_{50} - \text{temperature}}{\text{slope}}}\right)} \tag{18}$$

## Circulating variants

On 23 July 2025, the circulating-variants dataset allprot0723 was downloaded directly from the Alignment & Proteins section of the GISAID EpiCoV database. NSP3 protein sequences were identified with seqkit grep [44] by searching FASTA headers for the tag "NSP3". The boundaries of the PLpro were then mapped with seqkit locate [44]:

start motif: EVRTIKVFTTVDNINLHTQVVDMSMTYGQQ (≤ 2 mismatches)

end motif: DGALLTKSSEYKGPITDVFYKENSYTTTIK (≤ 2 mismatches)

The coordinates returned by these searches were used to excise the PLpro region from each NSP3 sequence. A total of 17,192,683 PLpro variants were identified. The corresponding EPI_ISL accessions were uploaded to GISAID, generating the fixed reference set EPI_SET_250801st (DOI: 10.55876/gis8.250801st).

Sequences whose lengths differed from 315 aa were excluded, as they likely contain insertions or deletions. Amino acid substitutions in PLpro were called with the customized find_mismatch function deposited (code ocean DOI: 10.24433/CO.0426220.v1) [18], using the Wuhan sequence (GeneBank access number: NC_045512.2) as the reference.

## Statistical analysis

DMS fitness scores were calculated using DiMSum [29]. All other subsequent calculations were performed in R studio with key parameters specified above. All experiments were performed at least twice independently, unless specified. Data are presented as the means ± SD, as indicated in the figure legends.

## Supporting information

**S1 Fig. Structure of PLpro.** A-B) structural representation of PLpro bound to LRGG, which represents the final four residues at the C-terminus of ubiquitin (PDB: 6XAA) [14]. A) highlights domains: the ubiquitin-like domain in grey, thumb domain in cyan, palm domain in red, and fingers domain in orange. The LRGG substrate is depicted in blue, with its key interacting residues shown in stick representation. B) presents PLpro in a surface view, highlighting the S4 pocket, blocking loop, active site and the narrow channel leading to the active site.
(TIFF)

**S2 Fig. Structures of PLpro binding to GRL0617 and 3k.** A-B) structural representation of PLpro bound to three inhibitors GRL0617 (PDB: 7CJM, PLpro in purple) [16], and 3k (PDB: 7TZJ, PLpro in green) [15]. The central panel zooms in on these compounds and their surroundings, highlighting key interaction partners. The right panel displays the structural formulae of these compounds.
(TIFF)

**S3 Fig. FACs gating strategy for cellular drug screening.** Live cells are identified using forward-scatter (FSC-A) and side scatter (SSC-A) parameters. Single cells were isolated by comparing FSC-A and FSC-H. FRET positive cells were detected using mRuby3 fluorescence (YG610/20) and FRET (B610/20).
(TIFF)

**S4 Fig. Sequence-function map of PF-07957472 escape.** This sequence-function map shows normalized DiMSum fitness scores calculated from FRET+ versus FRET- gates after inhibitor treatment. Variants are arranged with residue number on the x-axis and mutation type on the y-axis. Fitness scores were normalized so that the mean of wildtype and nonsense variants (1–305) are 1 and 0 respectively. The color scale represents the normalized fitness scores for each variant. Each square corresponds to a single-residue substitution and includes an inset slash whose length is proportional to the estimated error. Wildtype residues are highlighted with a solid circle, and variants with an error greater than 0.5 are marked with a cross.
(TIFF)

**S5 Fig. Sequence-function map of Jun12682 escape.** This sequence-function map shows normalized DiMSum fitness scores calculated from FRET+ versus FRET- gates after inhibitor treatment. Variants are arranged with residue number on the x-axis and mutation type on the y-axis. Fitness scores were normalized so that the mean of wildtype and nonsense variants (1–305) are 1 and 0 respectively. The color scale represents the normalized fitness scores for each variant. Each square corresponds to a single-residue substitution and includes an inset slash whose length is proportional to the estimated error. Wildtype residues are highlighted with a solid circle, and variants with an error greater than 0.5 are marked with a cross.
(TIFF)

**S6 Fig. Sequence-function map of WEHI-P8 escape.** This sequence-function map shows normalized DiMSum fitness scores calculated from FRET+ versus FRET- gates after inhibitor treatment. Variants are arranged with residue number on the x-axis and mutation type on the y-axis. Fitness scores were normalized so that the mean of wildtype and nonsense variants (1–305) are 1 and 0 respectively. The color scale represents the normalized fitness scores for each variant. Each square corresponds to a single-residue substitution and includes an inset slash whose length is proportional to the estimated error. Wildtype residues are highlighted with a solid circle, and variants with an error greater than 0.5 are marked with a cross.
(TIFF)

**S7 Fig. Sequence-function map of leaky expression.** This sequence-function map shows normalized DiMSum fitness scores calculated from FRET+ versus FRET- gates prior to induction of PLpro expression by doxycycline. Variants are arranged with residue number on the x-axis and mutation type on the y-axis. Fitness scores were normalized so that the

mean of wildtype and nonsense variants (1–305) are 1 and 0 respectively. The color scale represents the normalized fitness scores for each variant. Each square corresponds to a single-residue substitution and includes an inset slash whose length is proportional to the estimated error. Wildtype residues are highlighted with a solid circle, and variants with an error greater than 0.5 are marked with a cross.
(TIFF)

**S8 Fig. Variants that escape from PF-07957472.** The sequence-function map displays Z-scores that have been corrected based on each variant's activity and leaky expression scores and indicate the escape of each variant from inhibition. This data is also plotted in a different format in Fig 2B. Variants are arranged with residue number on the x-axis and mutation type on the y-axis. The color scale represents the Z-scores for each variant, with Z-scores less than 1 appearing white and Z-scores of more than 1 increasingly red. Each square corresponds to a single-residue substitution. Wildtype residues are highlighted with a solid circle. Variants that meet the filtering criteria from S20 Fig and have Z-scores higher than 2 are highlighted with boxes and used in the Venn diagrams depicted in Fig 4F and 4G. Variants with Z-scores over 2 but have low underlying counts or high errors in the contributing datasets are marked with a X. More information on these criteria can be found in the Methods section and S20 Fig.
(TIFF)

**S9 Fig. Variants that escape from Jun12682.** The sequence-function map displays Z-scores that have been corrected based on each variant's activity and leaky expression scores and indicate the escape of each variant from inhibition. This data is also plotted in a different format in Fig 2B. Variants are arranged with residue number on the x-axis and mutation type on the y-axis. The color scale represents the Z-scores for each variant, with Z-scores less than 1 appearing white and Z-scores of more than 1 increasingly red. Each square corresponds to a single-residue substitution. Wildtype residues are highlighted with a solid circle. Variants that meet the filtering criteria from S20 Fig and have Z-scores higher than 2 are highlighted with boxes and used in the Venn diagrams depicted in Fig 4F and 4G. Variants with Z-scores over 2 but have low underlying counts or high errors in the contributing datasets are marked with a X. More information on these criteria can be found in the Methods section and S20 Fig.
(TIFF)

**S10 Fig. Variants that escape from WEHI-P8.** The sequence-function map displays Z-scores that have been corrected based on each variant's activity and leaky expression scores and indicate the escape of each variant from inhibition. This data is also plotted in a different format in Fig 2B. Variants are arranged with residue number on the x-axis and mutation type on the y-axis. The color scale represents the Z-scores for each variant, with Z-scores less than 1 appearing white and Z-scores of more than 1 increasingly red. Each square corresponds to a single-residue substitution. Wildtype residues are highlighted with a solid circle. Variants that meet the filtering criteria from S20 Fig and have Z-scores higher than 2 are highlighted with boxes and used in the Venn diagrams depicted in Fig 4F and 4G. Variants with Z-scores over 2 but have low underlying counts or high errors in the contributing datasets are marked with a X. More information on these criteria can be found in the Methods section and S20 Fig.
(TIFF)

**S11 Fig. SDS-PAGE analysis of recombinant PLpro variants.** Recombinantly produced PLpro variants were run by SDS-PAGE to determine purity and relative concentration. The SeeBlue Plus2 pre-stained protein marker was used for molecular weight determination. The band with the expected molecular weight of PLpro is marked with an arrow. Each lane shows the respective PLpro variant as labeled, while the arrow indicates PLpro band.
(TIFF)

**S12 Fig. Native mass spectrometry analysis of PLpro variants.** A) Deconvoluted spectra from native MS experiment are shown for selected PLpro variants. The primary peak – representing the variant's mass – is highlighted with a

dashed line and normalized to 100% intensity for clarity. B) The predicted versus observed mass shift of each variant from wildtype.
(TIFF)

**S13 Fig. Representative dose-response curves for PLpro variants treated with PF-07957472.** In each panel, the emission from buffer-only control (no PLpro) was normalized 0, and the emission from samples without inhibitor was normalized to 100. Inhibitor concentrations are plotted on a logarithmic scale on the x-axis, with normalized emission on the y-axis.
(TIFF)

**S14 Fig. Representative dose-response curves for PLpro variants treated with Jun12682.** In each panel, the emission from buffer-only control (no PLpro) was normalized 0, and the emission from samples without inhibitor was normalized to 100. Inhibitor concentrations are plotted on a logarithmic scale on the x-axis, with normalized emission on the y-axis.
(TIFF)

**S15 Fig. Representative dose-response curves for PLpro variants treated with WEHI-P8.** In each panel, the emission from buffer-only control (no PLpro) was normalized 0, and the emission from samples without inhibitor was normalized to 100. Inhibitor concentrations are plotted on a logarithmic scale on the x-axis, with normalized emission on the y-axis.
(TIFF)

**S16 Fig. Kinetic analysis of PLpro variants during R-LRGG-AMC cleavage.** AMC fluorescence was measured every 5 minutes for 3 hours. Each dot is an individual measurement, while the curve represents the mean of three technical replicates from one representative experiment. Similar results are observed across two independent biological replicates.
(TIFF)

**S17 Fig. Kinetic analysis of PLpro variants during Ub-Rh110Gly cleavage.** Rh110Gly fluorescence was measured every 5 minutes for 3 hours. Each dot is an individual measurement, while the curve represents the mean of three technical replicates from one representative experiment. Similar results are observed across two independent biological replicates.
(TIFF)

**S18 Fig. Kinetic analysis of PLpro variants during ISG15-Rh110Gly cleavage.** Rh110Gly fluorescence was measured every 5 minutes for 3 hours. Each dot is an individual measurement, while the curve represents the mean of three technical replicates from one representative experiment. Similar results are observed across two independent biological replicates.
(TIFF)

**S19 Fig. Activity scores of key residues.** This sequence-function map snippet is of previously reported activity scores [18]. Variants are arranged with residue number on the x-axis and mutation type on the y-axis. The color scale represents the fitness scores for each variant. The means of wildtype and nonsense variants (residue 1–305) are normalized to 1 and 0, respectively. Each square corresponds to a single-residue substitution and includes an inset slash whose length is proportional to the estimated error. Wildtype residues are highlighted with a solid circle. Variants tested in our recombinant protein assays are boxed.
(TIFF)

**S20 Fig. Filtering of unreliable data.** A) The distribution of variants with their errors from leaky dataset (top) and activity dataset (bottom) in the absence of inhibition. A cutoff at 1.0 was chosen based on the distribution profile and used to exclude variants from Venn diagrams in Fig 4F and 4G and mark variants with an X in S8–S10 Figs. B-D) The error (top) and mean

count (bottom) distribution of PF-07957472 (B), Jun12682 (C), and WEHI-P8 (D). Cutoffs at 0.8 (error) and 10 (mean count) were used to exclude variants from Venn diagrams in Fig 4F and 4G and mark variants with an X in S8–S10 Figs.
(TIFF)

**S21 Fig. Map of variants achievable by a single base-pair edit.** Wildtype PLpro sequence from the Wuhan strain (Genebank access number: NC_045512.2) was used in the analysis. Variants are arranged with residue number as x-coordinates and mutation type as y-coordination. Variants able to be achieved via a single base-pair edit are shown in green. Variants accessible by more than one base-pair mutation are shown in grey.
(TIFF)

**S22 Fig. Heatmap for drug 3k/5c DMS dataset of key residues.** The data is downloaded from our previous publication [18]. Variants are arranged with residue number on the x-axis and mutation type on the y-axis. The color scale (shown in the color bar) represents the fitness scores for each variant. Each square corresponds to a single-residue substitution and includes an inset slash whose length is proportional to the estimated error. Wildtype residues are highlighted with a solid circle. The means of wildtype and nonsense variants (residue 1–305) are normalized to 0 and 1 respectively to enable easier interpretation. Those analyzed in the current study are boxed.
(TIFF)

**S1 Table. Escape variants.** Escape variants identified from DMS screening. The 2nd column displays all confident escape variants, the 3rd column highlights the subset of variants achievable via a single base-pair edit.
(XLSX)

**S2 Table. Properties of escape variants.** Escape variants for each compound are shown with associated Z-score, activity fitness score and associated error, whether a single base pair mutation can achieve the variant, and the number of times they were observed in GISAID data.
(XLSX)

**S1 Dataset. Variant data.** Data used to calculate Z-scores by variant, select escape variants and normalized fitness scores by variant. Data are arranged by variant and labeled either with mutation, or if synonymous wild-type, by DNA sequence (rows). Columns are labeled by treatment, with activity referring to PLpro activity after doxycycline induction, leaky referring to activity in the absence of doxycycline, PF, Jun and WEHI referring to PLpro activity in the presence of PF-07957472, Jun12682 and WEHI-P8 respectively. The sheet labeled "Raw Data - Z-Score calculation" outlines how raw DiMSum fitness scores were converted to Z-scores and gives the mean read count of the input condition (FRET+) and the DiMSum error estimations. Also included are GISAID observations and whether each variant is accessible by a single base pair mutation. The sheet labeled "Confident escape variants" provides context for how putative escape variants were filtered based on mean read counts (must be over 10) and DiMSum errors (must be below 1 for activity and leaky errors and below 0.8 for inhibitor treatments). Z-scores over 2 (2 SD above wildtype) were considered escape variants. The "conclusion" column labels whether a variant is categorized as escape (escape) or susceptible (sensitive) to a particular treatment or undetermined due to poor data quality (fail). Also included are GISAID observations and whether each variant is accessible by a single base pair mutation. The sheet labeled "Normalized fitness and SE" contains normalized DiMSum fitness scores by variant calculated from Raw Fitness scores that can be found in the sheet labeled "Raw data - Z-score calculation".
(XLSX)

## Acknowledgments

We gratefully acknowledge all data contributors, i.e., the authors and their originating laboratories responsible for obtaining the specimens, and their submitting laboratories for generating the genetic sequence and metadata and sharing via the GISAID Initiative, on which this research is based. The DMS libraries were created with the help of the WEHI

Multiplexed Assay Technology Hub. The authors thank Lianju Shen, Vineet Vaibhav and Amanda Bergamin of WEHI proteomics for their support and assistance in this work. The authors gratefully acknowledge WEHI Flow Cytometry, and the WEHI Advanced Genomics Facility as well as the WEHI National Drug Discovery Centre for technical resources and advice. We thank Hengyi Pacific Pty Ltd for their donation to support COVID-19 research; and a donation from AWM Electrical to support Australian drug discovery research to the National Drug Discovery Centre. This work was supported by computational resources provided by WEHI HPC, Milton. Work in the laboratories of the authors was made possible through Victorian State Government Operational Infrastructure Support (OIS) and the Australian Government NHMRC Independent Research Institute Infrastructure Support (IRIIS) Scheme.

## Author contributions

**Conceptualization:** Matthew E. Call, Melissa Joy Call.

**Data curation:** Xinyu Wu.

**Formal analysis:** Xinyu Wu, Melissa Joy Call.

**Funding acquisition:** Guillaume Lessene, David Komander, Melissa Joy Call.

**Investigation:** Xinyu Wu.

**Methodology:** Xinyu Wu, Melissa Joy Call.

**Project administration:** Shane M. Devine, Matthew E. Call, Melissa Joy Call.

**Resources:** Margareta Go, Julie V. Nguyen, Bernadine G. C. Lu, Katie Loi, Nathan W. Kuchel, Kym N. Lowes, Jeffrey P. Mitchell.

**Software:** Xinyu Wu.

**Supervision:** David Komander, Matthew E. Call, Melissa Joy Call.

**Validation:** Xinyu Wu, Melissa Joy Call.

**Visualization:** Xinyu Wu, Melissa Joy Call.

**Writing – original draft:** Xinyu Wu.

**Writing – review & editing:** Xinyu Wu, Shane M. Devine, Margareta Go, Julie V. Nguyen, Guillaume Lessene, David Komander, Matthew E. Call, Melissa Joy Call.

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
