## [Decision Letter · Decision Letter 0]

10 Jul 2025

PPATHOGENS-D-25-01080

Divergent resistance pathways amongst SARS-CoV-2 PLpro inhibitors highlight the need for scaffold diversity

PLOS Pathogens

Dear Dr. Call,

Thank you for submitting your manuscript to PLOS Pathogens. After careful consideration, we feel that it has merit but does not fully meet PLOS Pathogens's publication criteria as it currently stands. Therefore, we invite you to submit a revised version of the manuscript that addresses the points raised during the review process.

Please submit your revised manuscript within 30 days Sep 08 2025 11:59PM. If you will need more time than this to complete your revisions, please reply to this message or contact the journal office at plospathogens@plos.org. Please include the following items when submitting your revised manuscript:

We look forward to receiving your revised manuscript.

Kind regards,

Richard J. Kuhn, PhD

Academic Editor

PLOS Pathogens

Michael Letko

Section Editor

PLOS Pathogens

Sumita Bhaduri-McIntosh

Editor-in-Chief

PLOS Pathogens

orcid.org/0000-0003-2946-9497

Michael Malim

Editor-in-Chief

PLOS Pathogens

orcid.org/0000-0002-7699-2064

**Additional Editor Comments:**

Your manuscript was reviewed by experts in the field and they found it to be impactful and well executed. There are a few minor comments that should be addressed prior to acceptance, especially from reviewer #2.

**Journal Requirements:**

At this stage, the following Authors/Authors require contributions: Xinyu Wu, Shane M. Devine, Margareta Go, Julie V. Nguyen, Bernadine G. C. Lu, Katie Loi, Nathan W. Kuchel, Kym N. Lowes, Jeffrey P. Mitchell, Guillaume Lessene, David Komander, Matthew E. Call, and Melissa Joy Call. Please ensure that the full contributions of each author are acknowledged in the "Add/Edit/Remove Authors" section of our submission form.

- ® on page: 31.

4) Please ensure that the funders and grant numbers match between the Financial Disclosure field and the Funding Information tab in your submission form. Note that the funders must be provided in the same order in both places as well. State what role the funders took in the study. If the funders had no role in your study, please state: "The funders had no role in study design, data collection and analysis, decision to publish, or preparation of the manuscript.".

**Reviewers' Comments:**

Reviewer's Responses to Questions

**Part I - Summary**

Reviewer #1: Wu, Call and colleagues present an in depth analysis of drug resistance potential in PLPro from SARS-CoV-2 towards three inhibitors with nanomolar potency. A careful mutational scanning approach was employed to measure the impacts of almost all point mutations in PLPro on cleavage of a FRET substrate using flow cytometry to separate functional and non-functional populations and sequencing to evaluate the function of each mutation in a mixed pool. The authors identified mutations that can cause resistance to all three inhibitors. Two of the inhibitors shared the same scaffold and these showed a high degree of overlap in resistance mutations, consistent with the principle that inhibitor diversity can be a valuable approach for mitigating drug resistance potential.

Reviewer #2: This study explores the potential for drug resistance in SARS-CoV-2 papain-like protease (PLpro) inhibitors, with an emphasis on cross-resistance among structurally related compounds. Using deep mutational scanning, the authors evaluated resistance profiles for two potent GRL0617-derived inhibitors—PF-07957472 and Jun12682—as well as WEHI-P8, a structurally distinct inhibitor that binds to a similar site. Despite their shared target, PF-07957472 and Jun12682 showed largely overlapping escape mutations, reflecting their similar chemical scaffolds and binding modes. In contrast, resistance mutations to WEHI-P8 were distinct, suggesting that structurally diverse inhibitors are less likely to be compromised by shared resistance pathways. These findings highlight the importance of incorporating chemical diversity into PLpro inhibitor development to mitigate cross-resistance and improve antiviral durability.

**Part II – Major Issues: Key Experiments Required for Acceptance**

Reviewer #1: Viral infections and drug resistance have a tremendous impact on human health. This is an important study because it carefully examines drug resistance potential in a viral pathogen. Furthermore, the manuscript was very well written making it easy to follow and the experiments all appear to have been caried out with careful planning and execution. It was a pleasure to read. I have but one question for the authors – and it may be something that I missed. Since the study includes a large number of mutational measurements, it may be worthwhile to consider a some sort of approach to account for multiple tests (FDR, Bonferroni, etc). It is clear to me from the data that this is in no way going to change the conclusions in a meaningful way, and if not already implemented would make improve the statistical analyses. This is a minor issue and in no way detracts from my enthusiastic support of publication of this work in PLoS Pathogens.

Reviewer #2: The drug resistance of Jun12682 and PF-07957472 was recently reported (listed below). The authors are encouraged to compare the results from the current study and the results in the following paper.

Tan, H., Zhang, Q., Georgiou, K. et al. Identification of naturally occurring drug-resistant mutations of SARS-CoV-2 papain-like protease. Nat Commun 16, 4548 (2025).

One critical aspect of drug resistance is the impact of mutations on viral replication fitness. Although the authors have not validated the identified PLpro mutations using recombinant viruses, a detailed discussion of the enzymatic activity of these mutants—and how they compare to the wild-type enzyme—is warranted. Such analysis would provide important insights into the functional consequences of resistance mutations and their potential to emerge and persist in circulating viral populations. More importantly, the authors should also discuss which PLpro mutations are most likely to arise naturally based on mutation frequency, structural constraints, and evolutionary conservation.

Can the cellular FRET biosensor assay detect PLpro variants carrying double or triple mutations? It is well established that resistant viruses selected through serial viral passage often harbor multiple mutations. Furthermore, have the authors evaluated the enzymatic activity or resistance profile of any double or triple mutants derived from the identified single mutations? Such analysis would provide a more comprehensive understanding of potential resistance pathways.

**Part III – Minor Issues: Editorial and Data Presentation Modifications**

Reviewer #1: (No Response)

Reviewer #2: (No Response)

PLOS authors have the option to publish the peer review history of their article (what does this mean? ). If published, this will include your full peer review and any attached files.

**Do you want your identity to be public for this peer review?** For information about this choice, including consent withdrawal, please see our Privacy Policy .

Reviewer #1: No

Reviewer #2: No

**Figure resubmission:**
---

## [Editor Report · Decision Letter 1]

19 Aug 2025

Dear A/Prof Call,

We are pleased to inform you that your manuscript 'Divergent resistance pathways amongst SARS-CoV-2 PLpro inhibitors highlight the need for scaffold diversity' has been provisionally accepted for publication in PLOS Pathogens.

Best regards,

Richard J. Kuhn, PhD

Academic Editor

PLOS Pathogens

Michael Letko

Section Editor

PLOS Pathogens

Sumita Bhaduri-McIntosh

Editor-in-Chief

PLOS Pathogens

orcid.org/0000-0003-2946-9497

Michael Malim

Editor-in-Chief

PLOS Pathogens

orcid.org/0000-0002-7699-2064

The authors have addressed the minor issues that were raised by the reviewers. The manuscript is now suitable for publication and represents an important contribution for understanding resistance pathways against PLpro inhibitors of SARS-CoV-2.
---

## [Editor Report · Acceptance letter]

Dear A/Prof Call,

We are delighted to inform you that your manuscript, " 

Divergent resistance pathways amongst SARS-CoV-2 PLpro inhibitors highlight the need for scaffold diversity," has been formally accepted for publication in PLOS Pathogens.

Best regards,

Sumita Bhaduri-McIntosh

Editor-in-Chief

PLOS Pathogens

orcid.org/0000-0003-2946-9497

Michael Malim

Editor-in-Chief

PLOS Pathogens

orcid.org/0000-0002-7699-2064